# Experiences of a Regional Quality Improvement Collaborative to Reduce Unplanned Extubations in the Neonatal Intensive Care Unit

**DOI:** 10.3390/children9081180

**Published:** 2022-08-07

**Authors:** Melissa U. Nelson, Joaquim M. B. Pinheiro, Bushra Afzal, Jeffrey M. Meyers

**Affiliations:** 1Division of Neonatology, Department of Pediatrics, Crouse Hospital, Syracuse, NY 13210, USA; 2Department of Pediatrics, State University of New York Upstate Medical University, Syracuse, NY 13210, USA; 3Division of Neonatology, Department of Pediatrics, Albany Medical College, Albany, NY 12208, USA; 4Division of Neonatology, Department of Pediatrics, University at Buffalo, Buffalo, NY 14203, USA; 5Division of Neonatology, Department of Pediatrics, Harvard University School of Medicine, Boston, MA 02115, USA; 6Division of Neonatology, Department of Pediatrics, University of Rochester Medical Center, Rochester, NY 14642, USA

**Keywords:** quality improvement, collaborative, unplanned extubations, neonates, airway safety

## Abstract

Background: Unplanned extubations (UEs) occur frequently in the neonatal intensive care unit (NICU). These events can be associated with serious short-term and long-term morbidities and increased healthcare costs. Most quality improvement (QI) initiatives focused on UE prevention have concentrated efforts within individual NICUs. Methods: We formed a regional QI collaborative involving the four regional perinatal center (RPC) NICUs in upstate New York to reduce UEs. The collaborative promoted shared learning and targeted interventions specific to UE classification at each center. Results: There were 1167 UEs overall during the four-year project. Following implementation of one or more PDSA cycles, the combined UE rate decreased by 32% from 3.7 to 2.5 per 100 ventilator days across the collaborative. A special cause variation was observed for the subtype of UEs involving removed endotracheal tubes (rETTs), but not for dislodged endotracheal tubes (dETTs). The center-specific UE rates varied; only two centers observed significant improvement. Conclusions: A collaborative approach promoted knowledge sharing and fostered an overall improvement, although the individual centers’ successes varied. Frequent communication and shared learning experiences benefited all the participants, but local care practices and varying degrees of QI experience affected each center’s ability to successfully implement potentially better practices to prevent UEs.

## 1. Introduction

Unplanned extubation (UE), defined broadly as any extubation that was not previously intended for that time or not performed electively [1], occurs frequently among neonates admitted to the neonatal intensive care unit (NICU). UEs have been reported as the fourth most common adverse event in North American NICUs and the most common adverse event specifically associated with mechanical ventilation [2]. A systematic review of UEs in the NICU reported UE rates ranging between 0.14 to 5.3 UEs per 100 ventilator days [3]. Patients who experience a UE can suffer subsequent acute decompensation that requires cardiopulmonary resuscitation [4,5], emergent reintubation, and increased oxygen requirement [4,6,7]. Adverse events associated with intubation are more common with urgent intubation following UEs rather than intubations for other reasons [8]. UEs in preterm infants are also associated with worse long-term outcomes, including a longer duration of mechanical ventilation and a higher frequency of chronic lung disease, in addition to an increased NICU length of stay and increased hospital costs [9].

Given the potential short- and long-term consequences of UEs and the associated increased healthcare costs, UE prevention has become an important focus of quality improvement (QI) efforts for NICU patients. Because single-center QI initiatives have been successful in reducing UEs in the NICU [10,11,12,13,14,15,16,17,18,19,20,21] and one NICU among our regional perinatal centers (RPC) had been working to improve UE locally without much success, we proposed a multi-center QI collaborative as a model to drive improvement.

We, thus, formed a regional QI collaborative involving the four RPC Level IV NICUs in upstate New York to jointly perform a prospective observational QI project targeting the reduction of UE rates. The baseline monthly combined UE rate across the four RPCs was 4.1 per 100 ventilator days, and the project aim was to reduce the UE rates at each of the RPCs below an initial benchmark of two UEs per 100 ventilator days with a stretch goal of less than one UE per 100 ventilator days.

## 2. Materials and Methods

### 2.1. Collaborative Creation and Context

Representatives from each of the RPCs in upstate NY (Albany, Buffalo, Rochester, and Syracuse) formed a QI collaborative in 2014 with the shared goal of reducing UEs at their centers and for the entire region as a whole. These four RPCs, which are similar-sized NICUs in relatively close geographical proximity, have a long-standing history of collaboration. Each RPC provides the highest level of care to preterm and critically ill newborns in their respective regions of upstate NY, including all pediatric subspecialty and surgical services. The Level IV NICUs at each center have 52 to 68 beds and collectively have approximately 4000 NICU admissions per year.

Each RPC developed its own multidisciplinary team focused on local efforts to reduce UEs. Two centers already had teams and systems in place focusing on UE reduction, and two centers created new teams at the outset of the collaborative. The teams were comprised of attending neonatologists, fellows in training, advanced-practice providers, nurses, and respiratory therapists, whom all brought their shared knowledge and perspectives together to address UE reduction.

### 2.2. Ethical Approval

The proposals for this collaborative QI project were submitted for review to the institutional review boards (IRBs) as required at each center. The IRBs determined that the proposed activities were not considered to be research involving human subjects as defined by the Department of Health and Human Services and, therefore, IRB review and approval were not required.

### 2.3. Operational Definitions

Before gathering the baseline data, an operational definition was needed to ensure the consistent tracking of UEs across the centers. We chose to define UE using an inclusive definition proposed previously [1]. A UE was defined as any extubation that was not previously intended for that time or not performed electively. Each UE was then further classified based on whether the endotracheal tube (ETT) was considered dislodged or removed. We classified UEs as being due to a dislodged ETT (dETT) when there was evidence to directly support the dislodgment of the ETT outside of the trachea (e.g., obvious on inspection, confirmation with laryngoscopic visualization, and audible crying); otherwise a UE was classified as a removed ETT (rETT) when ETT removal occurred without clear evidence of ETT dislodgment (e.g., tube found secured without evidence of displacement and removed due to concern for the tube’s obstruction with or without evidence of irreversible tube obstruction).

### 2.4. Outcome Measures

The primary outcome measure was the monthly combined UE incidence rate of the four RPCs, with the UE incidence rate defined as the number of UE events per 100 ventilator days. Each RPC reported the number of ventilator days and the total number of UEs, from which a combined UE incidence rate was calculated per month. All of the UEs were also classified as either dETT or rETT events, and the incidence rates of dETT and rETT UEs per 100 ventilator days were also calculated on a monthly basis at each RPC, and the mean rates for the four RPCs determined in an identical fashion to the combined UE incidence rates. Statistical process control charts (QI Macros, KnowWare International, Inc., Denver, CO, USA) were used to display the outcome data over time. The special cause variation (the rule of shift) was determined using suggested criteria specific to healthcare [22].

Additional secondary outcome measure data was collected for each UE event to help identify the potential risk factors associated with UEs. A detailed description of each UE was provided by the involved staff members immediately following the UE occurrence, using standardized forms to facilitate the apparent cause analysis. Additional information of interest included patient demographics, NICU census and staffing, the location and timing of the event, the associated care activities, and the subsequent outcome, such as the need for resuscitation, reintubation, or a change in respiratory support.

### 2.5. Data Collection and Reporting

Each RPC developed a reliable method of data collection and standardized reporting to track the UEs. Every UE event that occurred in the NICU was counted based on the collaborative’s inclusive operational definition. Data collection tools were created at each RPC to foster accurate and thorough data collection. Data were collected at each center during a six-month initial observational period from August 2014 to February 2015 to establish a baseline for each NICU and the collaborative as a whole prior to any interventions. Summarized UE incidence rate data from each RPC was shared on a monthly basis. Secondary outcome de-identified data from all four RPCs was compiled into a collective database (REDCap) to catalog data elements and details associated with each UE occurrence.

### 2.6. Benchmarking Goals

After assessing our baseline data, we chose a benchmarking goal of less than two UEs per 100 ventilator days as potentially achievable. We adopted a stretch goal of less than one UE per 100 ventilator days, which was previously suggested as a benchmark for NICUs in a systematic review [3] and subsequently demonstrated to be feasible [10].

### 2.7. Regional Collaboration and Interventions

The regional collaborative held monthly conference calls to compare the UE incidence rates, share observations and experiences, and ultimately help generate improvement strategies. Frequent communication via email enabled the further sharing of ideas and materials. Annual in-person regional conferences and site visits were conducted to foster collaboration between teams and visualize interventions in real-time.

After the baseline UE incidences were determined, the multidisciplinary teams at each RPC reviewed their own local data and developed key driver diagrams to identify local trends and potential risk factors associated with UEs (Figure 1). This information was utilized to develop center-specific tests of change to reduce UEs. Each center then performed one or more PDSA (Plan-Do-Study-Act) cycles that were designed to address either dETT-specific events or rETT-specific events and, in some instances, both (Table 1).

In early 2015, each RPC held unit-wide educational sessions for all NICU staff to discuss the risks associated with UEs, review the published evidence on the successful reduction of UEs through QI efforts, and provide the rationale for forming the collaborative as a structure for improvement. The educational sessions included a lecture component, question-and-answer segment, and hands-on practice with ETT securement techniques utilizing mannequins.

All RPCs performed a PDSA cycle with the implementation of a bedside airway card (Figure 2) for every intubated patient to quickly convey information to the healthcare team during routine patient care as well as during the real-time assessment of potential UEs. The airway cards were customized to meet each center’s practices and placed at the bedside of each intubated patient to provide specific information, including the ETT size, location of securement, chronological notations regarding the ETT position adjustment, and comments on unique airway considerations. The airway cards also served as a visual cue for airway safety and helped to ensure consistency in the documentation and assessment of the ETT.

Another test of change adopted by all centers involved standardizing the approach for evaluating an intubated infant with acute respiratory decompensation to avoid unnecessary rETT-specific UEs. Though they varied slightly in the level of detail, each center used the mnemonic “ABCD,” which stood for Auscultate/Assess breath sounds, Bag/provide manual breaths with higher pressure, Check end-tidal CO2, and Direct visualization with laryngoscopy. Standardizing the approach ensured that the NICU staff would perform several critical steps prior to performing an rETT to ensure that ETT removal was the best course of action. The mnemonic and algorithm for “ABCD” was also incorporated into the bedside airway cards as an additional visual reminder, as exemplified in Figure 2. RPC D reinforced the application of the “ABCD” algorithm through multidisciplinary simulation exercises at an annual skills fair.

All RPCs standardized ETT securement practices within their own centers to address dETT-specific UEs by utilizing commercial ETT securement devices, testing different taping techniques (an ETT secured at the midline versus the corner of mouth), and/or trialing several different types of adhesive tape. RPCs A and D adopted the use of the NEO-fit^TM^ ETT holder (CooperSurgical, Inc., Trumbull, CT, USA), whereas RPC C transitioned from routinely using tape to secure the ETTs to the use of the NeoBar^®^ ETT holder (Neotech, Valencia, CA, USA). Positional aids and devices (hats, mittens, beanbags, and bumpers) were utilized to discourage patient movements that could result in UEs. Three RPCs (A, B, and C) adopted new policies for staff members’ roles in ETT adjustment and securement (e.g., two people are required to move the intubated patient or re-secure the ETT). RPCs B and C created small groups of staff to act as an “airway task force” to conduct frequent audits and provide real-time feedback on ETT securement practices.

The trends observed during the project led to the implementation of subsequent PDSA interventions. RPC C developed an airway risk scoring system based on the observation that many patients experienced two or more UEs. The scoring system utilized a green, yellow, and red scale to signify a UE risk (green = baseline risk, yellow = increased risk due to previous UE, and red = high risk due to critical airway status, such as severe micrognathia or subglottic stenosis), and color-coded alerts were added to the bedside airway cards for additional point-of-care visual reminders.

## 3. Results

### 3.1. Primary Outcome

A total of 1167 UEs were reported by the four RPC NICUs. RPC A abandoned the collaborative after 2 years due to time and staffing constraints, and the three other RPCs continued joint QI work for an additional 2 years. The baseline monthly combined UE rate was 4.1 per 100 ventilator days; the UE rate varied from 2.3 to 6.2 per 100 ventilator days at each RPC (Figure 3). A special cause variation was observed beginning in February 2017 when the combined UE rate decreased from 3.7 to 2.5 per 100 ventilator days, representing a 32% reduction. The clinical impact of this reduction equates to approximately 100 fewer UEs per year across the collaborative. Although each RPC witnessed improvement, the overall reduction in UEs was driven by larger reductions at two of the four RPCs (Figure 4).

Further analysis of UEs based on classification as dETT versus rETT showed that the majority of UEs at all the RPCs were classified as dETT (N = 941, 81%) versus rETT (N = 226, 19%). Following the implementation of one or more PDSA cycles, no appreciable change in the combined rate of dETTs was noted (Figure 5). No special cause variation was observed for the rate of dETTs; however, the rate fell below the baseline for ten of the last eleven months of the project. On the other hand, a substantial reduction in the combined rate of rETTs was noted with a special cause variation observed beginning in October 2016 when the combined rETT UE rate decreased from 0.9 to 0.3 per 100 ventilator days (67% reduction) (Figure 6).

### 3.2. Secondary Outcomes and Additional Observations

Additional descriptive data were collected regarding patient demographics and events associated with UEs, and several notable trends were observed. The majority of UEs occurred in premature patients of gestational age of fewer than 28 weeks and those with very low birth weights (VLBW, birth weight < 1500 g). Gender-related differences were noted, with UEs more common in male infants. Of patients who experienced a UE, between 22 to 41% experienced two or more UEs with variations between observations at each RPC.

A Pareto analysis of the activities associated with UEs revealed that several types of events occurred commonly (Figure 7). Hands-on patient care activities with the NICU staff was the most common associated event. UEs often occurred during periods of ETT securement or adjustment and when patients were very active and/or agitated. Subsequent Pareto analyses at each center guided the implementation of additional changes through new PDSA cycles, based on center-specific key driver diagrams, as exemplified in Figure 1.

The patient outcome following a UE event varied between the RPCs. At RPCs B and D, the majority of infants required re-intubation shortly after a UE, whereas at RPCs A and C, approximately half of the UEs resulted in the infant transitioning to non-invasive support or room air (Figure 8). If a patient required reintubation, it often took multiple attempts to successfully place the ETT, with variation between the RPCs of 21% to 45% of intubations requiring multiple attempts (Figure 9). Three percent of UEs were complicated by cardiorespiratory decompensation, requiring significant cardiopulmonary resuscitation (chest compressions +/− epinephrine). An ETT occlusion was also a rare phenomenon, observed in 1% of all UEs and 7% of rETTs.

## 4. Discussion

Our regional QI collaborative was successful in reducing the combined UE rate among Level IV NICUs across upstate New York. Although we did not achieve our original benchmark goal of less than two UEs per 100 ventilator days, we adopted a broad, inclusive definition of UE with an operational definition to ensure all possible events were captured. To our knowledge, this is the earliest collaborative designed to reduce UEs, and we achieved an overall reduction in UEs of more than 30%. The improvements were not uniform among the centers, and the most significant decline occurred in UEs classified as rETT. This suggests that a significant proportion of staff-assisted UEs can be avoided by the coordinated education of front-line caregivers and the implementation of standardized rapid evaluation algorithms for intubated infants who undergo acute respiratory decompensation.

The classification of UEs is novel, and this approach allows teams to design and test targeted interventions to address each UE type. For example, a trial of new ETT securement devices would be most impactful for dETT UEs, and the adoption of a standardized assessment algorithm for the decompensation of mechanically ventilated patients would be most impactful for rETT UEs. By classifying UEs into these two specific subtypes, we were able to focus our efforts according to the greatest need and then more accurately assess the stratified results for meaningful improvement. We were able to successfully reduce the rates of rETTs by 67% across the collaborative, most likely due to the adoption of the “ABCD” algorithm, which created a standardized approach to the assessment of a mechanically ventilated patient experiencing acute decompensation and provided a pause moment for staff to determine whether ETT removal was truly the best course of action. Consequently, we prevented many UEs that were avoidable, thus decreasing the harm and discomfort associated with physiologic destabilization and reintubation, even when ETTs are removed by staff. Our inclusive definition and classification scheme ensure that these events are not overlooked, as they might be when neonatologists exclude intentional but non-elective extubations [23], although an inclusive definition may result in significantly higher observed UE rates [24]. No significant reduction in the frequency of dETT UEs was observed, which may be a result of a lack of standardization on which tests of change to carry out across centers, differing levels of experience and expertise with the QI methodology, or other center differences in the processes that influence the rate of UEs not captured in our work.

Similar to the observations reported by others, we observed that the smallest [5,25] and most premature infants [5] were those who most frequently experienced UEs during our project. These patients have anatomical challenges and often require prolonged mechanical ventilation, which contributes to an increased likelihood of experiencing UE [5]. Consistent with the findings from others, the data from our collaborative is a reminder that UE prevention strategies should target this NICU subpopulation. We also observed a higher than the anticipated frequency of patients with multiple UEs, which has also been noted by others [5,7,9,26]. The recognition of previous UE occurrence as a potential risk factor for subsequent UE occurrence helps to increase awareness of NICU staff and gives an opportunity for extra attention and prevention strategies for these patients. Nevertheless, the majority of patients affected by UEs experience a single event; thus, UE reduction efforts must target the prevention of a first UE.

We observed several outcomes following UEs that were comparable to findings reported by other UE QI projects. Immediate reintubation was often required following UEs, occurring in 49% to 82% of the UE cases at each RPC. Other studies have reported similar outcomes, with between 58% to 76% of patients requiring reintubation following a UE [9,26,27,28]. Furthermore, we observed that reintubation often took multiple attempts, ranging between 21% to 45% of cases at each RPC. Since airway injury can occur during intubation, the high frequency of reintubation following a UE and multiple reintubation attempts to achieve success emphasizes why UE remains a large patient safety concern. Cardiovascular collapse and the need for cardiopulmonary resuscitation (CPR) have also been observed in conjunction with UEs. We observed a comparable frequency of the need for CPR (3%) following a UE when compared with others (3% to 13%) [9,26,27]. Although these events, fortunately, do not occur as frequently, this risk of serious morbidity further highlights the importance of UE prevention.

Collaborative QI projects in neonatology have been shown to improve clinical outcomes in the NICU and decrease morbidities, resource use, and length of stay, which all result in the reduction of healthcare costs [29]. Successful examples include the work of the Ohio Perinatal Quality Collaborative in reducing late-onset sepsis and improving outcomes for patients with neonatal narcotic abstinence syndrome [30,31], the reduction in length of stay by the California Perinatal Quality Care Collaborative [32], and numerous endeavors undertaken by the Vermont Oxford Network’s iNICQ QI collaboratives [33,34,35]. At the outset of our project, we hoped to employ similar regional collaborative QI strategies to address UEs in the NICU. More recently, the implementation of a UE prevention bundle at 43 centers as part of the Children’s Hospitals Solutions for Patient Safety network resulted in a 24% reduction in UEs, though the work was carried out among neonatal, pediatric, and cardiac intensive care units [27]. The UE prevention bundle involved three factors: standardized anatomic reference points and an ETT securement method, a protocol for moving high-risk intubated patients, and an apparent cause analysis following UEs. Similar to our collaborative, they achieved an overall reduction in UEs across the collaborative but had differing levels of success at individual centers. They observed the most significant reduction in UEs at centers with the highest and most sustained bundle compliance.

The collaboration with multiple RPCs offered many opportunities for shared learning but uncovered challenges during our efforts to reduce UEs. While there are many similarities between the four RPCs, each center has unique characteristics in terms of physical design and staffing models that made standardization challenging. For example, RPC C does not have fellows, and respiratory therapists do not attend deliveries at RPC D. These factors likely result in differences in the degree of experience of staff performing intubations and possibly in the consistency of ETT securement procedures. Additionally, at the outset of the collaborative, there was a large difference in each of the RPCs’ level of experience in tracking and attempting to prevent UEs with local QI efforts. Some RPCs were primed for rapid and sustained success with established UE prevention teams already in place, whereas others were more novice to QI and had to devote more effort upfront to develop core teams and implement systems for carrying out the project. Not surprisingly, the most significant improvements occurred at the RPCs with the more established UE prevention QI programs, and the reduction in UE rates at two of the RPCs with the most experience with QI accounted for the improvement effect for the collaborative as a whole. While there was a great benefit in the frequent dialogue amongst all four centers and the opportunity to learn from each other’s experiences, both in terms of pitfalls and successes, the implementation of interventions was not synchronized across the RPCs, and improvement was not achieved uniformly.

The limitations of this study include the heterogeneity and relative asynchrony of the interventions undertaken at each RPC rather than a standardized bundle of care. Although the project was designed intentionally with this in mind in order to address the specific needs of each NICU, the individualized approaches made collaborative efforts less robust and an analysis of the efficacy of specific interventions challenging. More specifically, the heterogeneity of ETT securement methods across the RPCs precluded the conclusions on the relative superiority of any method; however, it is likely that the standardization of practices within each RPC contributed to the decreased UE risk. The underlying differences in the staffing patterns and airway management functions of staff noted above may have contributed to the variation in our results, but our study was not designed to capture such information. Another weakness was the lack of patients’ family involvement in the study design, UE education sessions, and bedside interventions. NICU patients’ family members assume a large role in their children’s care, and strategic family involvement would have likely been very impactful. Future endeavors should seek to include patients’ families throughout all of the project stages. A further limitation of this study was the inability of all the centers to reliably capture UEs occurring in NICU patients when located outside of the NICU-proper (e.g., in the delivery room, operating room, in-hospital transit, or inter-facility transport); however, these events accounted for 11% of UEs at the RPC with the most fastidious reporting methods, which would not account for the differences in the results among the centers. Finally, one other limitation of this project and any project seeking to reduce UEs within NICUs is the lack of a standard definition for UE [1,23,24]. Although our collaborative intentionally adopted a broad, inclusive definition shared by all participating RPCs at the outset of the project, the likely differences between our definition and the definition utilized by other institutions make benchmark comparisons difficult.

## 5. Conclusions

Our regional QI collaborative successfully reduced the combined UE rate across our region, but individual center outcomes varied. Frequent communication, data transparency, and shared learning promoted improvement efforts, but local care practices and the level of QI experience affected the degree of success with the implementation of potentially useful interventions at each center.

## Figures and Tables

**Figure 1 children-09-01180-f001:**
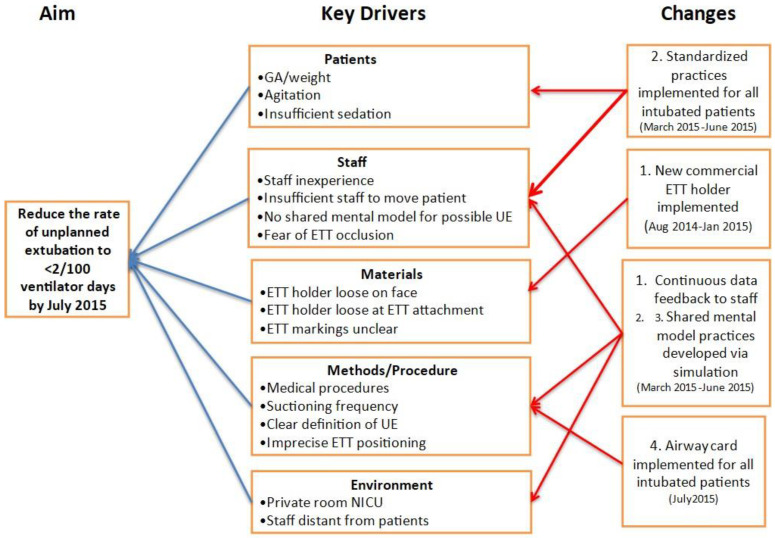
Example key driver diagram from Center D. Numbers under “Changes” indicate PDSA cycles.

**Figure 2 children-09-01180-f002:**
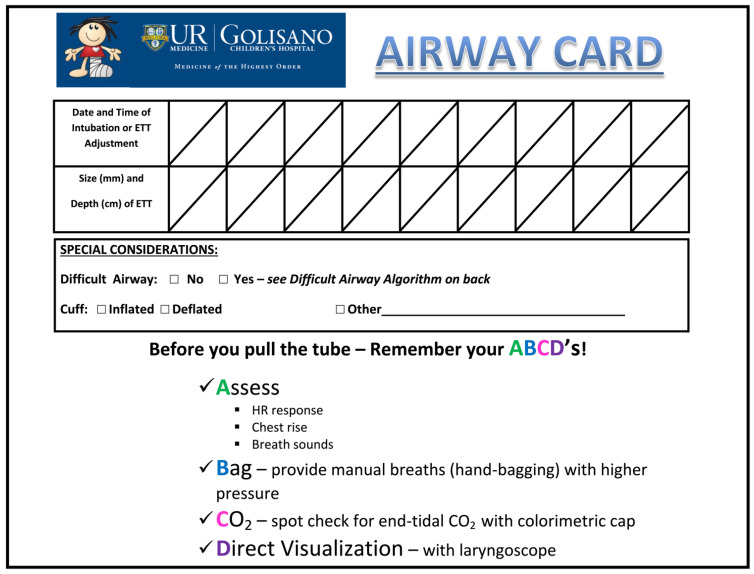
Example airway card.

**Figure 3 children-09-01180-f003:**
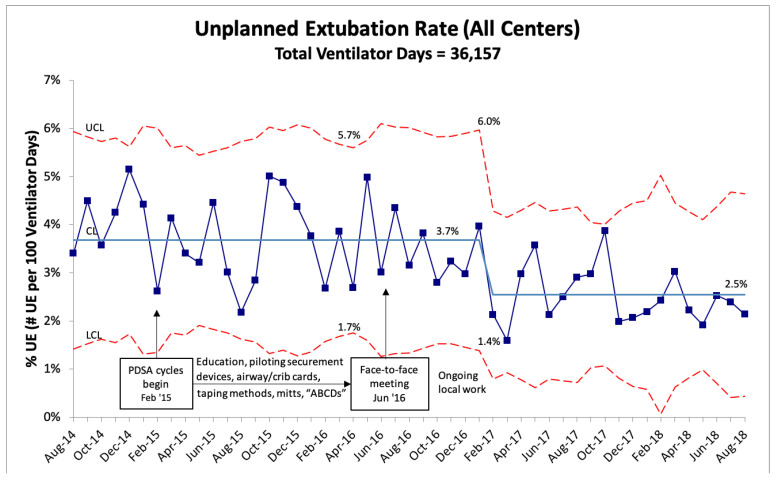
U-chart demonstrating the monthly combined UE rate for all centers. Special cause variation (rule of shift) was noted beginning in February 2017. CL = center line (mean), UCL = upper control limit, and LCL = lower control limit.

**Figure 4 children-09-01180-f004:**
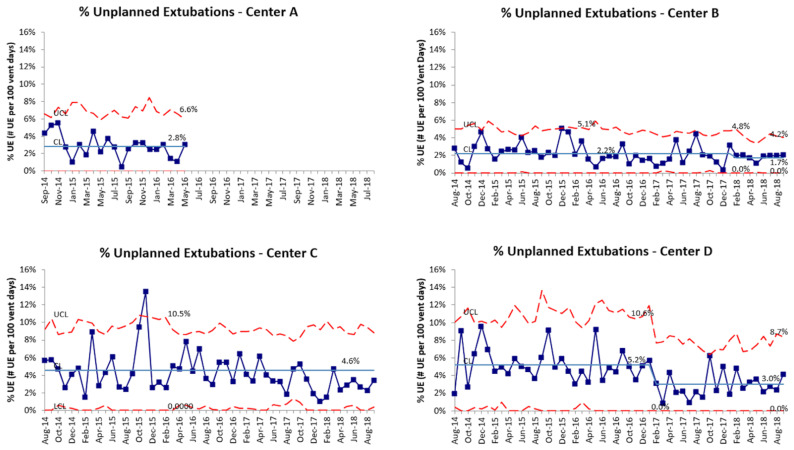
U-charts displaying the monthly UE rate for individual centers. Special cause variation (rule of shift) was noted at centers B and D only. CL = center line (mean), UCL = upper control limit, and LCL = lower control limit.

**Figure 5 children-09-01180-f005:**
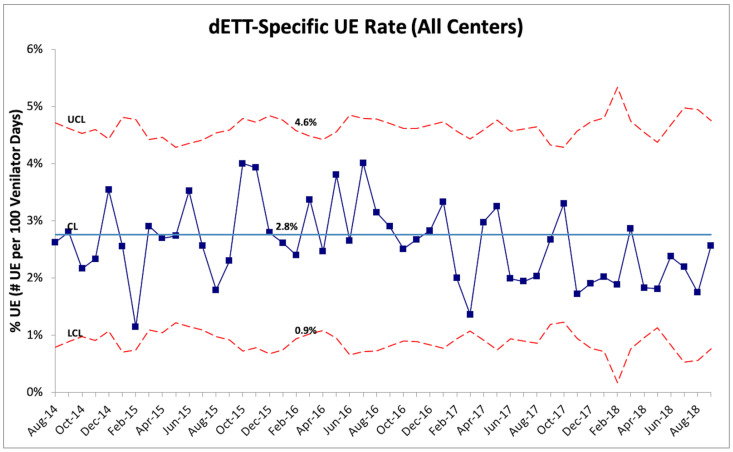
U-chart displaying the monthly UE rate for events classified as resulting from a dislodged ETT (dETT). No special cause variation was observed, though 10 of the last 11 months were noted below the baseline mean. CL = center line (mean), UCL = upper control limit, LCL = lower control limit, and ETT = endotracheal tube.

**Figure 6 children-09-01180-f006:**
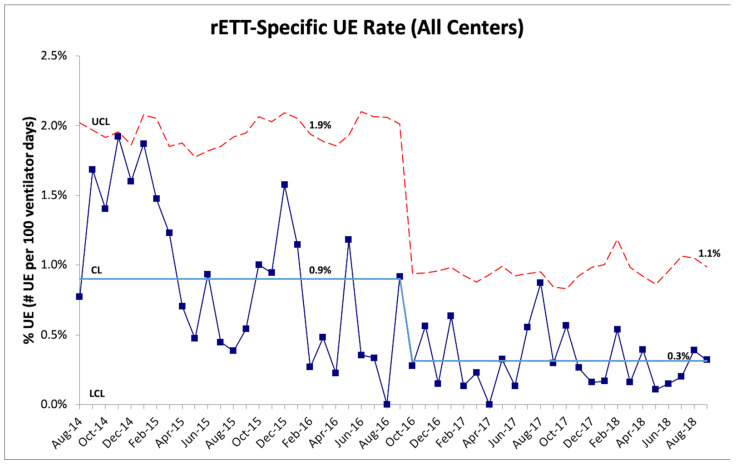
U-chart displaying the combined monthly UE rate for events classified as resulting from a removed ETT (rETT). Special cause variation (rule of shift) was noted beginning in October 2016. CL = center line (mean), UCL = upper control limit, LCL = lower control limit, and ETT = endotracheal tube.

**Figure 7 children-09-01180-f007:**
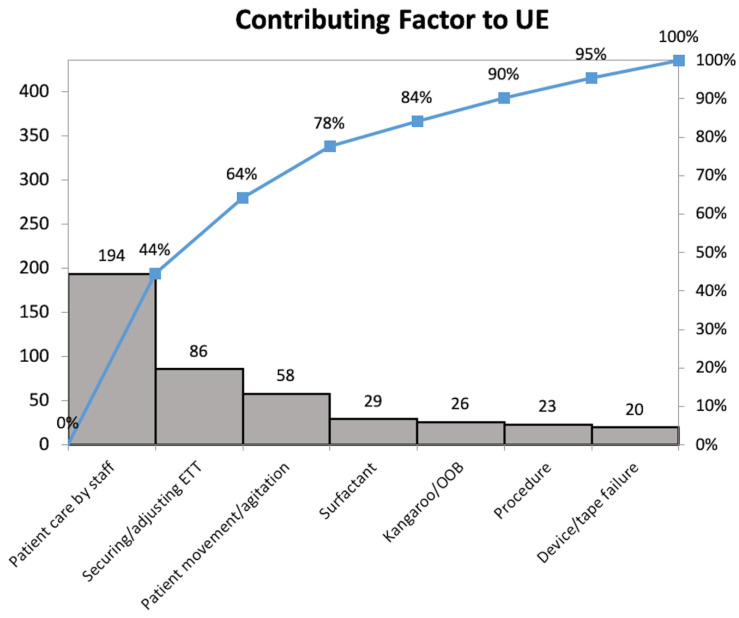
Pareto chart demonstrating the relative contribution of various factors for UEs among all four centers for the first 18 months of the collaborative. ETT = endotracheal tube and OOB = out of bed.

**Figure 8 children-09-01180-f008:**
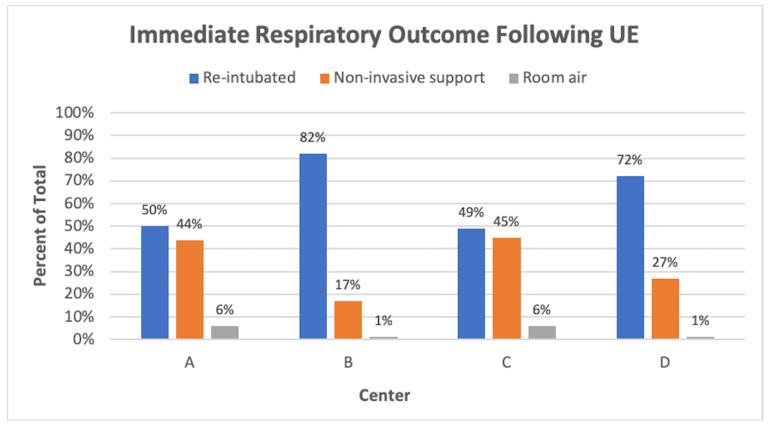
Percent of UEs that resulted in immediate reintubation, placement of infant on non-invasive respiratory support, or transition to no respiratory support (room air) at each center.

**Figure 9 children-09-01180-f009:**
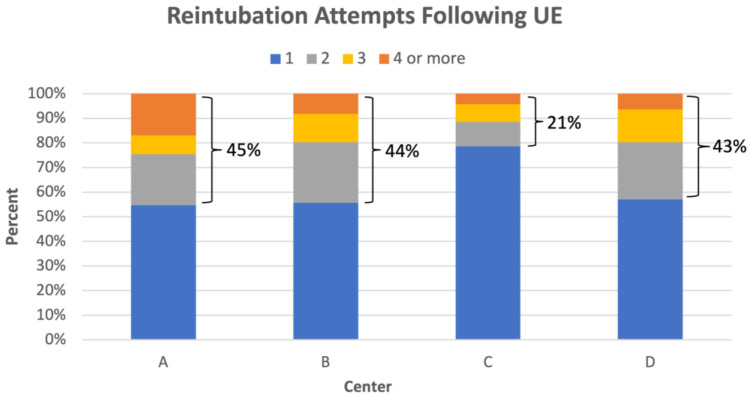
Percent distribution of number of intubation attempts following UEs at each center. The percentages bracketed beside each column represent the percentage of multiple intubation attempts.

**Table 1 children-09-01180-t001:** UE prevention interventions adopted by individual centers.

Intervention	Center A	Center B	Center C	Center D
Establishment of UE QI team	++	+	++	++
Adoption of UE operational definition	++	++	++	++
Standardized UE subtypes	++	++	++	++
Apparent cause analysis of UE events		++	++	
NICU-wide multidisciplinary education sessions on UE	++	+	++	++
Feedback and UE data sharing with staff	++	+	++	+
Bedside cards as visual reminders	++	++	++	++
Standardized “ABCD” approach to assess infants with desaturation/bradycardia to prevent unnecessary rETT	++	+, ++	++	++
Adoption of new ETT securement methods (commercial device, brand of tape, and taping strategy)	++	+, ++	++	++
Potential UE scenario simulations for staff education		+		+
Changes in staffing protocols for ETT adjustment and moving patients	++	++	++	+
Family education on UE prevention *				
Creation of “airway task force” to assess ETT position and securement		++	++	+
Extension of UE monitoring to NICU patients located outside of NICU (delivery room, operating room, and transport)				+

(+) = existed prior to collaborative. (++) = added through the collaborative. * Not implemented at any center. UE = unplanned extubation, QI = quality improvement, NICU = neonatal intensive care unit, ETT = endotracheal tube, and rETT = removed ETT.

## Data Availability

The data presented in this study are available in the text and figures of the manuscript. No other data is available for this QI work.

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
