# Peer review of "Experiences of a Regional Quality Improvement Collaborative to Reduce Unplanned Extubations in the Neonatal Intensive Care Unit"

_children, 2022, doi:10.3390/children9081180_

Round 1
Reviewer 1 Report
The manuscript by Nelson et al highlight quality improvemts efforts undertaken in 4 regional centers in zNew York state to handle unplanned extubations in NICU.
Authors May need to clarify following issues
introduction
Authors May want to write in their aim what baseline UE rates were and to what level they want to reduce to and by what percentage
Method
Benchmarking goals: authors may want to write what percent decrease in rates of UE
Figure 1
numbering at changes brackets in not clear
what are shared mental model practices?
Also authors may need to describe what information is in airway cards
It’s is not clear from manuscript or figures that what odds cycles were implanted by which units at what time point. Figure showing split information of all centers may show that information
Authors may want to say ABCD approach is for rETT only
There were only 4 RPCs. Rather than using some RPCs or several RPCs aothors should say which RPCs did what changes. That may help readers understand what worked on which places
Results
why did one rpc abandon in 2 years? Which rpc was that?
Is it because they already had low rates?
Did the authors look at the babies who had UE were they more likely to have endotracheal tube adjustments?
Author Response
Point 1: Introduction - Authors may want to write in their aim what baseline UE rates were and to what level they want to reduce to and by what percentage.
Response 1: We appreciate this suggestion and have added the baseline combined monthly UE rate to the project aim statement along with the UE rate reduction target and stretch goal (lines 54-57).
Point 2: Methods - Benchmarking goals: authors may want to write what percent decrease in rates of UE.
Response 2: Our UE rate reduction target of less than two UEs per 100 ventilator days was based on previously suggested benchmarking goals (Silva et al., Merkel et al.) rather than a percentage reduction. We have otherwise modified the introduction aim statement per this reviewer’s above suggestion.
Point 3: Figure 1 - numbering at changes brackets in not clear.
Response 3: We agree that this should have been clarified. The numbers under “Changes” indicate PDSA cycles. This has been updated in the Figure 1 legend.
Point 4: Figure 1 - what are shared mental model practices?
Response 4: The shared mental model is a common understanding by multidisciplinary team members of the steps to be taken during emergency evaluation of the ETT status, leading to coordinated execution of those steps. It relates to the practical implementation of the “ABCD” algorithm, which was slightly different at each center, and whose details are too extensive for this manuscript. Some additional details have been added to the manuscript for better clarification regarding the “ABCD” algorithm (lines 164-171).
Point 5: Also authors may need to describe what information is in airway cards.
Response 5: Modifications have been made to the methods section to provide further clarification regarding airway card content (lines 153-160) and an airway card example has been included as an additional figure (Figure 2 in new version of manuscript).
Point 6: It’s not clear from manuscript or figures that what odds cycles were implanted by which units at what time point. Figure showing split information of all centers may show that information
Response 6: We agree that this is the biggest limitation of our work and try to highlight our awareness of that limitation in the discussion (lines 356-360). Each RPC performed multiple individualized and asynchronous PDSA cycles throughout the project, which we tried to demonstrate in Table 1. Additional details have been added throughout the entire methods section as well for further clarification regarding which efforts were adopted by which RPCs.
Point 7: Authors may want to say ABCD approach is for rETT only.
Response 7: We updated the manuscript to more clearly state that that “ABCD” algorithm intervention applies specifically to rETT (lines 164-171).
Point 8: There were only 4 RPCs. Rather than using some RPCs or several RPCs authors should say which RPCs did what changes. That may help readers understand what worked on which places.
Response 8: We appreciate this suggestion and as stated above, the methods section has been updated in order to provide more clarification regarding which RPCs adopted which interventions.
Point 9: Results - Why did one rpc abandon in 2 years? Which rpc was that? Is it because they already had low rates?
Response 9: RPC A left the project early due to lack of time and staffing. We updated the manuscript accordingly to make the rationale behind their departure clearer (lines 196-198).
Point 10: Did the authors look at the babies who had UE were they more likely to have endotracheal tube adjustments?
Response 10: Yes, ETT adjustment was the second most common event associated with UE as demonstrated in the Pareto chart (Figure 7 in the updated manuscript) and text (lines 242-243).

Reviewer 2 Report
This is a well-written manuscript. Several quality improvement projects on the same topic have been published in the last four years, (see references, two of which were cited in the manuscript as well). It is clearly a topic that continues to be of interest to most neonatologists.
While the effort is duly noted, in the current form, it does not add to existing knowledge or to what recent manuscripts have described. It would be more appreciated by, and helpful for readers to know more details of interventions utilized in order to reduce the rate of unplanned extubations and improve outcomes.
For instance, in this manuscript, the authors have mentioned ‘commercial device, brand of tape, taping strategy’, but no specifics. Some of the other studies have described those specifics which are helpful for readers in both academic and practice settings. The key driver diagram mentions an ‘ETT holder’, which is not mentioned anywhere else. Were all the units consistent with the commercial device, brand of tape, or taping strategy? Or did everyone utilize different strategies and still have better outcomes anyway because they were more cautious to prevent unplanned extubations? More details about education sessions, crib cards, etc, rather than leaving it to the reader’s imagination would also be appreciated. Some other studies on a similar topic have also elaborated on adverse events associated with unplanned extubations. This manuscript has summarized some of them in small paragraphs #209 - #220, more details, even in a short table, would increase the scientific appeal of the manuscript.
Overall, it is a well-written manuscript, it has a unique perspective with the data representative of a regional quality improvement collaborative. With additional descriptive information, it would be advantageous and valuable to readers interested in quality improvement.
In Figure 8:
Did the authors mean Less than or equal to 2 and More than or equal to 3?
More than or equal to 2 and More than or equal to 3 seems contradictory.
References:
Bertoni, C. B., Bartman, T., Ryshen, G., Kuehne, B., Larouere, M., Thomas, L., ... & Moallem, M. (2020). A Quality Improvement Approach to Reduce Unplanned Extubation in the NICU While Avoiding Sedation and Restraints. Pediatric quality & safety, 5(5).
Kandil, S. B., Emerson, B. L., Hooper, M., Ciaburri, R., Bruno, C. J., Cummins, N., ... & Grossman, M. (2018). Reducing unplanned extubations across a children’s hospital using quality improvement methods. Pediatric quality & safety, 3(6).
Kim, F., Brooks, C., Villaraza-Morales, S., Daven, J., Bradley, S., Chhipa, A. K., ... & Vargas, D. (2021). Quality Improvement Effort to Decrease Unplanned Extubations in a Cardiac Neonatal Intensive Care Unit. Pediatric Quality & Safety, 6(Suppl 1).
Mahaseth, M., Woldt, E., Zajac, M. E., Mazzeo, B., Basirico, J., & Natarajan, G. (2020). Reducing unplanned extubations in a level IV neonatal intensive care unit: the elusive benchmark. Pediatric Quality & Safety, 5(6).
Author Response
Point 1: This is a well-written manuscript. Several quality improvement projects on the same topic have been published in the last four years, (see references, two of which were cited in the manuscript as well). It is clearly a topic that continues to be of interest to most neonatologists.
While the effort is duly noted, in the current form, it does not add to existing knowledge or to what recent manuscripts have described. It would be more appreciated by, and helpful for readers to know more details of interventions utilized in order to reduce the rate of unplanned extubations and improve outcomes.
Response 1: We appreciate the reviewer’s mention of the two additional previously uncited references (Kandil et al, Kim et al.). We were previously aware of both of these papers and projects but had not initially included them as references as they were not restricted to the general NICU (Kandil et al. focused efforts across an entire children’s hospital and Kim et al. focused efforts on CCU-NICU/general NICU combination). However, we recognize that their recent important work should be acknowledged, and therefore have also added these references to our manuscript.
Point 2: For instance, in this manuscript, the authors have mentioned ‘commercial device, brand of tape, taping strategy’, but no specifics. Some of the other studies have described those specifics which are helpful for readers in both academic and practice settings. The key driver diagram mentions an ‘ETT holder’, which is not mentioned anywhere else. Were all the units consistent with the commercial device, brand of tape, or taping strategy? Or did everyone utilize different strategies and still have better outcomes anyway because they were more cautious to prevent unplanned extubations?
Response 2: These are all excellent points and we agree that additional details would strengthen the manuscript. We have therefore included more details regarding commercial ETT holder devices and which RPCs utilized which device in the manuscript (lines 175-181) for further clarification. Each RPC attempted to standardize ETT securement practices within their own unit, but strategies were not standardized across the region. Analyzing the efficacy of any one particular intervention for airway securement was challenging given the heterogeneity of interventions. Anecdotally, none of the RPCs reported immediate or sustained success with one device over any other. The sentiment highlighted in this reviewer’s last question above seemed true – better outcome seemed more related to the standardization of ETT securement and extra caution for UE prevention within each RPC rather than due to one specific strategy or device. An additional statement has been added to the limitations section in the discussion to further highlight this issue (lines 360-363).
Point 3: More details about education sessions, crib cards, etc, rather than leaving it to the reader’s imagination would also be appreciated.
Response 3: We appreciate this suggestion as well. The entire methods section has been updated to provide more details regarding all interventions and an additional Figure has been added to provide an example of an airway card (Figure 2 in the updated manuscript).
Point 4: Some other studies on a similar topic have also elaborated on adverse events associated with unplanned extubations. This manuscript has summarized some of them in small paragraphs #209 - #220, more details, even in a short table, would increase the scientific appeal of the manuscript.
Response 4: Although we appreciate the suggestion, we do not have additional information regarding adverse events aside from what we already reported in the manuscript text and figures.
Point 5: In Figure 8: Did the authors mean Less than or equal to 2 and More than or equal to 3?
More than or equal to 2 and More than or equal to 3 seems contradictory.
Response 5: The data outcomes in the original Figure 8 were not contradictory but rather overlapping; however, this reviewer’s question highlighted an opportunity to make the results clearer. The manuscript text (lines 252-254, 313-314) and figure have been modified to show the results more clearly with a stacked bar graph (Figure 9 in the revised manuscript).

Reviewer 3 Report
This is overall an interesting manuscript on a surprisingly underrated topic. I fully appreciate the authors’ inclination towards this issue, the degree of involvement that they exhibit, and also the quality of the conclusions that they reach.
Some minor issues:
Rows 69-70, I suspect the teams were “comprised” not “compromised”
Rows 156-157: I especially applaud the implementation of the ABCD mnemonic, and I would suggest rather than keeping it in the text as it is now, the authors should place it in a separate cassette, in order for it to be more visible.
Rows 213-214 – the observation “between 22 to 41% experienced two or more UEs” is of particular interest and worthy of more insight, in my opinion, even if it was noted in similar studies.
Figure 1 – I’m not sure what “ETT holder loose on holder” in the Materials cassette means
The title for Figure 3 should be placed underneath the figure, on the previous page. Also, in Figure 3, I would be particularly curious to know what happened during the months Oct-Nov 2015 in center C, to have such a spike in unplanned extubations – maybe something about this should be mentioned in the Discussions section…
Figure 6 – the authors should explain the abbreviations they use: ETT, OOB
Figures 7 and 8 – the authors should also place tags with exact percentages for each column. Regarding the content of Figure 8, I am not sure it is vital to be illustrated…
Author Response
Point 1: Rows 69-70, I suspect the teams were “comprised” not “compromised”
Response 1: Thank you for catching this error, and the reviewer’s suspicion was correct. We have corrected the manuscript to state “comprised” (lines 70-71).
Point 2: Rows 156-157: I especially applaud the implementation of the ABCD mnemonic, and I would suggest rather than keeping it in the text as it is now, the authors should place it in a separate cassette, in order for it to be more visible.
Response 2: Thank you for appreciating our mnemonic as we were proud of it as well. We have modified the text to include additional details regarding the “ABCD” algorithm and also included an example of an airway card to show how it was utilized visually (Figure 2 in revised manuscript).
Point 3: Rows 213-214 – the observation “between 22 to 41% experienced two or more UEs” is of particular interest and worthy of more insight, in my opinion, even if it was noted in similar studies.
Response 3: We also found this observation particularly intriguing. Additional details have been added to the manuscript in the Methods section regarding an intervention undertaken by RPC C due to this observation (lines 187-193).
Point 4: Figure 1 – I’m not sure what “ETT holder loose on holder” in the Materials cassette means
Response 4: This phrase referred to looseness of the ETT holder relative to the point of attachment to the ETT as opposed to looseness at the point of attachment to the baby’s face. Figure 1 was updated accordingly to provide better clarity.
Point 5: The title for Figure 3 should be placed underneath the figure, on the previous page. Also, in Figure 3, I would be particularly curious to know what happened during the months Oct-Nov 2015 in center C, to have such a spike in unplanned extubations – maybe something about this should be mentioned in the Discussions section…
Response 5: The figure title is now placed correctly, beneath the figure (Figure 4 in the revised manuscript). Unfortunately, there was not a clear explanation for why Center C had the large spike in UEs during the two-month period between Oct-Nov 2015. There were no obvious red flags noted at that time (normal census, normal staffing ratios) and no recent/concurrent PDSA cycles or protocol changes to attribute to the increase.
Point 6: Figure 6 – the authors should explain the abbreviations they use: ETT, OOB
Response 6: Thank you for catching this error as well. Abbreviation explanations have been added to the figure legend (Figure 7 in the revised manuscript).
Point 7: Figures 7 and 8 – the authors should also place tags with exact percentages for each column. Regarding the content of Figure 8, I am not sure it is vital to be illustrated…
Response 7: Percentage tags have been added to these two figures and the last figure was changed to a stacked bar graph format to better convey results (Figures 8 and 9 in revised manuscript).

Round 2
Reviewer 2 Report
Thank you for the detailed responses, comments, and updates to the manuscript. These updates provide improved clarity to the main points.
May I please recommend a minor revision to this manuscript that has the potential to be very impactful? In the final discussion, you could you please highlight and summarize differences in different levels of NICUs while carrying out interventions as well as assessing outcomes? For instance, center C has a significantly higher success rate of intubations on the first attempt in comparison to centers A, B, and D. Are there center-specific challenges or differences in both implementing certain interventions as well as getting appropriate responses to the intervention that may assist readers in designing quality improvement projects for their respective units or regional collaboratives?
That would be an extremely impactful contribution of this work compared to previous work in this space and set it apart from other published studies of a similar nature.
Author Response
Point 1: May I please recommend a minor revision to this manuscript that has the potential to be very impactful? In the final discussion, could you please highlight and summarize differences in different levels of NICUs while carrying out interventions as well as assessing outcomes? For instance, center C has a significantly higher success rate of intubations on the first attempt in comparison to centers A, B, and D. Are there center-specific challenges or differences in both implementing certain interventions as well as getting appropriate responses to the intervention that may assist readers in designing quality improvement projects for their respective units or regional collaboratives?
That would be an extremely impactful contribution of this work compared to previous work in this space and set it apart from other published studies of a similar nature.
Response 1: We appreciate the Reviewer’s suggestion. We also found the differences among centers interesting, although we cannot convincingly explain the reasons for those differences. This applies to both baseline UE rates and the apparently higher first-time intubation success in NICU C. Cross-sectional differences were an opportunity to identify potentially better practices, then discuss and examine practices within each center. Ultimately, only decrease in UE rates longitudinally within centers (and overall) would suggest improvement with individual tests of change.
Although all participating NICUs are level IV NICUs in large regional referral centers, some functional heterogeneity is likely. There are known differences among the NICUs in their experience with quality improvement work, staff and resources dedicated to QI, and the size and composition of the teams involved in this project, in particular. There were also differences in staffing; for example, the types and experience of house staff (all NICUs have residents, but NICU C does not have fellows), ratio of house staff to mid-level providers, presence of respiratory therapists in the delivery room, and nighttime in-house attending coverage. While each of these factors may have influenced the results we observed, we do not have sufficient details to discern which may have been significant. In Table 1, we compare what we thought were the most relevant pre-existing and newly implemented features at each of the centers.
We speculate that the higher reintubation success on the first attempt may be related to more experienced staff performing intubations at Center C, but we did not collect data to test this speculation, as intubation was not the focus of our project. We added brief references to this issue in the Discussion, but it was already implicit in our listing of limitations.
In the penultimate paragraph, we added:
For example, Center C does not have fellows, while in Center D, respiratory therapists do not attend deliveries. These factors likely result in differences in the experience of staff performing intubations, and possibly in the consistency of ETT securement procedures. (lines 343-346)
To the last paragraph of the Discussion:
The underlying differences in staffing patterns and airway management functions of staff noted above may have contributed to the variation in our results, but our study was not designed to capture such information. (lines 366-368)
